# Hydrochemical Characteristics and Formation Mechanism of Groundwater in Qingdao City, Shandong Province, China

**Shenting Gang** [1,2,3]**, Tao Jia** [2,3]**, Yinger Deng** [1,]*****, Liting Xing** [4] **and Shuai Gao** [2,3]

1   College of Environment and Civil Engineering, Chengdu University of Technology, Chengdu 610059, China; gangshenting@163.com
2   801 Institute of Hydrogeology and Engineering Geology, Shandong Provincial Bureau of Geology & Mineral Resources, Jinan 250013, China
3   Shandong Engineering Research Center for Environmental Protection and Remediation on Groundwater, Jinan 250013, China
4   School of Water Conservancy and Environment, University of Jinan, Jinan 250022, China
*   Correspondence: dengyinge@mail.cdut.edu.cn

**Abstract:** The coastal area is a major area of socio-economic development and the most active zone for human activities. With the rapid development of the economy and the strengthening of urban construction, the groundwater environmental problems in coastal areas are increasingly prominent. It is significant to investigate the groundwater chemical characteristics, hydrochemical types, and the factors that influence groundwater chemistry for water resources protection and groundwater resources development. In this paper, 48 groundwater samples and 5 surface water samples from the study area were analyzed with statistical analysis, Piper diagram, Gibbs diagram, mineral saturation index method, and the ionic scale factor, and explored the factors that influence groundwater chemistry. The modified Nemerow index method was also applied to evaluate the groundwater. The results show that the groundwater in the study area is neutral to weakly alkaline (average pH = 7.0~8.0). The groundwater chemical types are mainly $Cl \cdot SO_4$-Na and $SO_4 \cdot Cl$-$Ca \cdot Mg$. Hydrochemistry is mainly influenced by rock weathering and evaporative concentration. TDS was strongly correlated with TDS, Na+, $Mg^{2+}$, $K^+$, $Ca^{2+}$, $Cl^-$, $SO_4{}^{2-}$, and the saturation index showed a gradual increase along the groundwater drainage flow path, it indicates that the main groundwater ions originate from the dissolution of halite, sulfate, and carbonate. Combining GIS technology and the kriging spatial interpolation method, we obtained the current situation map of groundwater quality in Laoshan District. The groundwater quality is mainly III water, and the overall water quality is good; IV and V water is mainly distributed in the middle and lower reaches of rivers, distributed in a belt pattern along the coastline. It is mainly influenced by both the human factor and seawater intrusion. It is significant for the utilization of groundwater resources and the management of seawater intrusion in the Laoshan District. In addition, the proposed research ideas and methods provide a reference for the study of groundwater genesis in other coastal areas in the world.

**Keywords:** groundwater; hydrochemical characteristics; formation mechanism; water quality assessment; Qingdao city

## 1. Introduction

Groundwater has always been widely available and is now the most dependable source of freshwater supply in recent times [1]. As an area of interaction between terrestrial freshwater and marine, brackish water, coastal areas are an indispensable and important part of the global land–ocean water cycle. The shortage of freshwater resources and water environment pollution are the main factors limiting the development of coastal areas; moreover, they pose a serious threat to the safety of the groundwater environment. The global-scale seawater intrusion has caused widespread concern in the international community [2]. In recent decades, with the rapid economic growth and the increased disturbance of

human activities, the groundwater in the coastal zone of eastern China has been threatened by both the decrease in water quantity and the deterioration of water quality [3]. The seawater intrusion can trigger water scarcity, soil salinization, and significant crop yield reduction in coastal areas. Assessing these processes remains a major challenge, and therefore, further understanding of the groundwater chemistry characteristics and influencing factors in coastal areas appears essential [4]. According to the Ministry of Natural Resources of China in 2019, seawater intrusion is more serious in the coastal areas of the Bohai Sea and Yellow Sea in China [5], and severe seawater intrusion is mainly distributed in the coastal localities of Shandong, Liaoning, Hebei, and Jiangsu [6]. Normally, people attribute the increased groundwater mineralization in coastal areas to modern seawater intrusion, but dissolved filtration of stratigraphic salinity, evaporative concentrations, seawater intrusion mixing, or stronger water-rock interactions all contribute to increased groundwater mineralization during long-term evolution [7].

The water chemistry analysis method can better identify the main factors affecting groundwater changes. As an important reference to reflect the environmental quality condition of groundwater, the water chemistry characteristics have received extensive attention at home and abroad. Currently, the main research methods related to groundwater chemical characteristics and their influencing factors are mathematical statistics, graphical procedures, ion ratio method, and mineral saturation index method [8–12]. Beyond that, geochemical modeling of water-rock interaction processes is one of the most powerful tools [13,14]. A comprehensive study of the northern Huangqihai Basin (a typical endorheic basin), based on isotopic and hydrochemical methods, was conducted by Jin Jing et al. [15]. Their study showed that the hydrochemical evolution was mainly affected by rock weathering and also by cation exchange. Manure, sewage, and $NH_4$ fertilizers were identified to be the main sources of nitrate contamination precipitation, while industrial activities and synthetic $NO_3$ were unlikely to be the main sources of nitrate contamination in the study area. The hydrochemical characteristics of strontium in the groundwater system of Shijiazhuang City and its formation mechanism were studied by Li Duo et al. [16] using provenance analysis, factor correlation analysis, and runoff condition analysis. The results show that strontium originates from the dissolution of strontium-containing minerals in carbonate rock, sheet hemp rock, clastic rock, and granite in the Taihang Mountain area of the Hutuo River Basin. The enrichment and distribution of strontium are related to groundwater runoff conditions. Jeen et al. [17] review seawater intrusion studies in the western coastal aquifers of South Korea conducted over the past 20 years. They show that groundwater geochemistry is largely affected by mixing with seawater, cation exchange processes during seawater intrusion, artificial contamination, water-rock interactions, and redox processes. They suggest that more modeling, laboratory experiments, isotope sampling, and microbial community monitoring should be conducted in these coastal aquifers. El Bab et al. [18] evaluated the quality of groundwater in the coastal aquifers of the Gaza Strip. Their study showed that the quality of water depends on the distance from the coastline and that groundwater in large areas along the coastline is not suitable for human consumption. Spatio-temporal analyses of pre-monsoon and post-monsoon groundwater levels of two coastal aquifer systems (upper leaky confined and underlying confined) were carried out in Purba Medinipur District by Halder Subrata et al. [19]. Their research shows that considerable spatial and temporal variability was found in the seasonal groundwater levels of the two aquifers, a trend of seawater intrusion, and an urgent need for appropriate measures for groundwater protection. Other scholars, such as Dong Q, have conducted systematic studies on the characteristics of groundwater chemistry in the Hetao Plain, the lower Yangtze River alluvial plain, and the Sichuan basin, revealing the groundwater genesis and evolutionary processes [20–23].

The study area is located in the northeastern coastal region of China. The rivers are seasonal rivers with strong evaporation. Its main characteristics are short length, fast flow, and mostly direct flow into the sea. Most of the seawater intrusion in Qingdao was formed in the 1970s and was most serious in the medium of 1980s. By the end of the 1990s, it entered

a relatively stable development stage. The main reason is that the groundwater extraction near the invasion area was greatly reduced, and the precipitation increased compared with the 1980s, which led to the groundwater level rebounding to different degrees, and some of the funnels recovered, the momentum of the sea (salt) water invasion was curbed, and the invasion area retreated. In the year 2002, which was a particularly dry year, the groundwater level in some areas continued to decline, and the invasion area expanded again [24]. Since most of the seawater intrusion areas are economically developed and densely populated, the catastrophic nature of coastal seawater intrusion has become more obvious after entering the 21st century.

From the early 1970s to the present day, geological and other scientific departments have carried out research in the study area of geology, hydrogeology, and environmental geology. Scholars such as Wu Jichun, Zheng Xilai, and Cheng Jianmei have researched seawater intrusion simulation, forecasting, and prevention [25–29]. The relevant studies by Gautam A and other scholars provide experience and reference [4,30,31]. However, additional in-depth research is required to better understand the groundwater resource situation and to use and protect groundwater resources sustainably.

In this paper, we selected groundwater in Laoshan District, Qingdao City, as the research object. We analyzed the groundwater chemical characteristics and influencing factors by statistical analysis, Piper diagram, Gibbs diagram, ion ratio, and mineral saturation index, based on previous research results. The groundwater quality was also evaluated by the modified Nemerow index method, aiming to provide a scientific basis for the evolution of water chemistry and ecological and environmental protection in the study area and similar coastal areas.

## 2. Material and Methods

### 2.1. Overview of the Study Area

Qingdao sits on the northeastern coast of China, and the study area situates in the Laoshan District of Qingdao. The Study Area extends from the northern bank of the Tuzhai River to the north, which borders the coast of Jimo City; south to Maidao, which connects the coast of Shinnan District; east to the coastline; and westwards to a boundary extending 3–5 km inland from the coastline. The geographical coordinates are 36°3′9.3″~36°20′9.4″ N, 120°24′9.78″~120°39′24.6″ E (Figure 1). The study area mainly covers the middle and lower reaches of the coastal rivers (Tuzhai River, Wanggezhuang River, Xiaowang River, Liangshui River, Nanjiushui River, etc.) and the plain alluvial area in front of the mountains. Additionally, it includes part of the coastal aquaculture mudflat area, with an area of about 245 km$^2$.

The study area has a temperate monsoon climate with intense monsoons and significant marine precipitation. The exposed strata are relatively simple, from old to new: Metasedimentary, Mesozoic Cretaceous, and Cenozoic Quaternary strata. According to the study area's aquifer media types and storage conditions, there are three main types of groundwater: open rock-like pore water, clastic rock-like pore-fissure water, and bedrock fissure water [28] (Figure 2a). Groundwater comprises predominantly Quaternary pore-bearing rock formations, mainly distributed in the middle and lower river valley plains of the major and minor rivers in the study area. The permeability is relatively better, and the water's volume increases with the aquifer's thickness [24]. The water volume from a well varies between 100–500 m$^3$/d. Only in the area of Huangshan-Qingshan Village southeast of the study area can we find clastic rock-like pore-fissure water [24]. The lithology of the aquifer is mainly the feldspathic sandstone of the Lower Cretaceous Laiyang Formation [29]. The fractured structure is anisotropic, and the permeability and water-richness of the aquifer are weak. Water-rich small lots can form in lithological contact zones or low-lying areas of tectonic development, with water volumes between 50–100 m$^3$/d, suitable for decentralized rural water use. Groundwater is mainly recharged by atmospheric precipitation, and discharge is controlled by topography, which generally flows from west to east (Figure 2b).

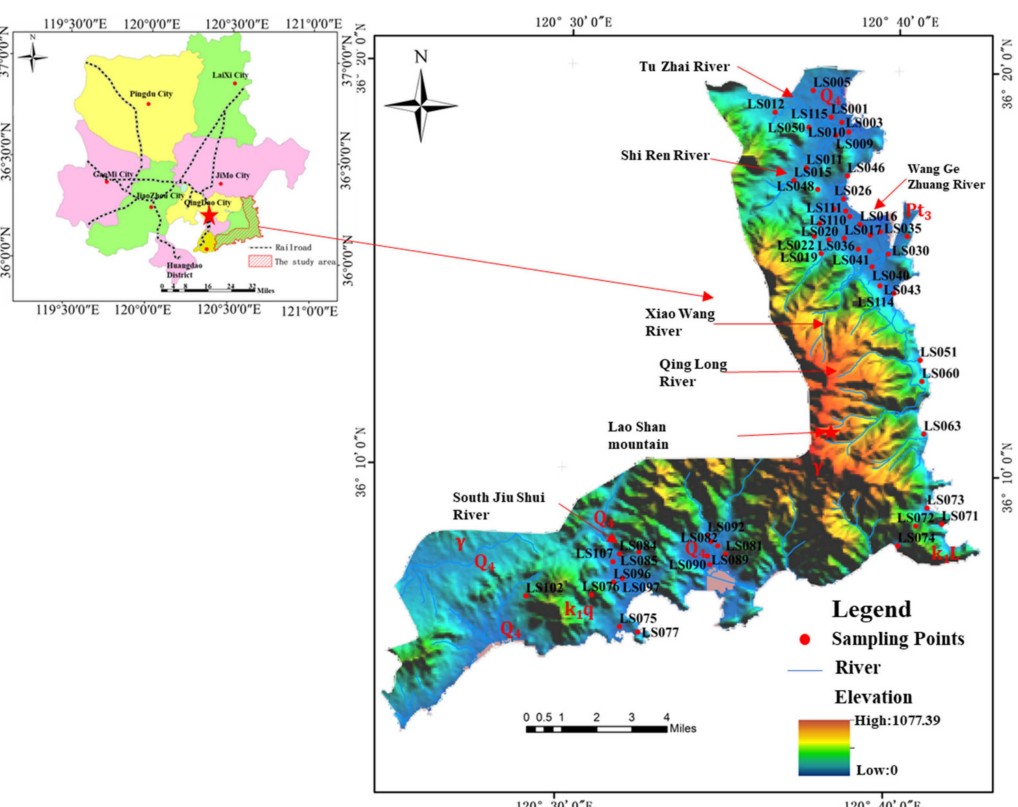

**Figure 1.** Location map of the study area.

### 2.2. Sample Collection and Processing

The study collected 48 sets of groundwater samples from December 2016 to September 2017, including 29 sets of Quaternary pore water (LS001, LS003, LS005, LS009, LS012, LS015, LS016, LS024, LS026, LS027, LS029, LS030, LS033, LS035, LS040, LS041, LS043, LS046, LS048, LS073, LS074, LS075, LS083, LS084, LS089, LS090, LS092, LS097, and LS104) and 19 sets of bedrock fracture water (LS010, LS011, LS020, LS022, LS023, LS051, LS060, LS063, LS071, LS076, LS077, LS081, LS082, LS085, LS096, LS102, LS111, LS110, and LS019), and five sets of surface water, based on the analysis of hydrogeological conditions and topographic features. Pore water and fracture were determined according to the water-bearing lithology, groundwater dynamic conditions, and well depth where the investigated water wells were located. All water samples were tested and analyzed in the laboratory of the Shandong Province Geological and Mining Engineering Survey Institute. $K^+$, $Na^+$, $Ca^{2+}$, $Mg^{2+}$, $NH_4^+$, $HCO_3^-$, $Cl^-$, $SO_4^{2-}$, $NO_3^-$, TDS, TA, TH, pH, and oxygen consumption were determined. The inorganic cations ($K^+$, $Na^+$, $Ca^{2+}$, $Mg^{2+}$) were determined using an OPTIMA 7000DV ICP-OES from Perkin Elmer Co. The limits of detection were 0.07 mg/L, 0.03 mg/L, 0.02 mg/L, and 0.02 mg/L. The anions ($Cl^-$, $SO_4^{2-}$, $NO_3^-$) were determined by CIC-D120 ion chromatograph from Qingdao Shenghan Chromatography Technology Co. The limits of detection were 0.007 mg/L, 0.018 mg/L, and 0.016 mg/L. $HCO_3^-$ was determined using a buret from Tianbo Glass Instruments Co. Total alkalinity is the total amount of carbonate and bicarbonate, and oxygen consumption COD is calculated by the $COD_{Mn}$ method with $O_2$

The samples were collected, preserved, and transported with reference to the Technical Specification for Groundwater Environmental Monitoring (HJ/T 164-2004) [32]. The detection method and basis refer to GB/T8538-2016 [33] and GB/T5750-2006 [34]. GPS was used to locate the geographical position of the sampling point, and the location of the sampling point is shown in Figure 1.

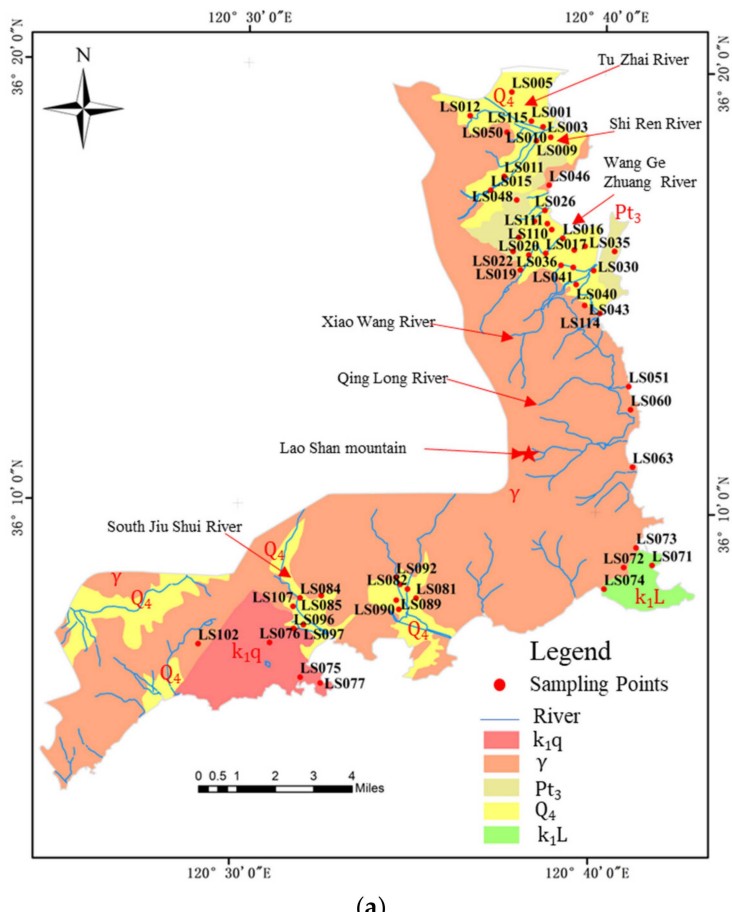

(**a**)

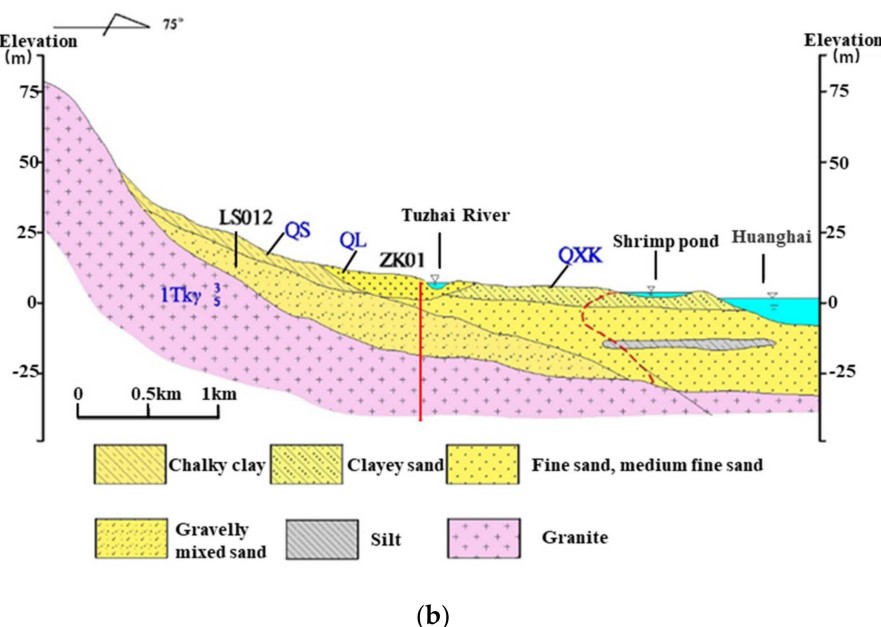

(**b**)

**Figure 2.** Schematic diagram of the hydrogeological conditions in the study area. (**a**) Distribution of aquifer media types (**b**) Schematic hydrogeological cross-section of the Tuzhai River, Laoshan District, Qingdao (West–East).

*2.3. Analytical Research Methods*

The distribution of sampling points in the study area was schematically drawn using ArcGIS 10.1. We used Origin2017 to conduct descriptive statistical analysis of water chemistry parameters and plotted Piper trilinear plots, Gibbs plots, and ion ratio plots.

*2.4. Modified NEMEROW Index Method of the Comprehensive Evaluation*

2.4.1. NEMEROW Index Method

The NEMEROW index method is the method recommended in the national standard (GB/T14848-2017) [35]. This method has the advantages that other comprehensive evaluation methods do not have, such as a straightforward mathematical process and convenient operation. The maximum score for the evaluation parameter highlights the impact of the grossly exceeded indicators on the evaluation results. However, there are issues with the comprehensive evaluation of groundwater: (i) The $F_i$ values are discrete, describing groundwater quality discontinuities, and cannot accurately reflect groundwater quality status. (ii) Groundwater was evaluated in an integrated manner giving too much prominence to the impact of $F_{max}$. The role of $F_{ave}$ was not evident, as demonstrated by Kou Wenjie et al. [36].

2.4.2. Modified NEMEROW Index Method

① Modification of $F_i$. In the NEMEROW index method, the $F_i$ value is based on the groundwater quality category classification, and the $F_i$ is calculated according to Equation (1). This technique avoids the problem of the discontinuous values of $F_i$ taken in the NEMEROW index method.

$$F_i = \begin{cases} 0 & (j = 1, u_i \leq S_{ij}) \\ \frac{S_{ij} - u_i}{S_{ij} - S_{i(j+1)}} & (j = 1, S_{ij} \leq u_i \leq S_{i(j+1)}) \\ 1 + 2 \times \frac{S_{ij} - u_i}{S_{ij} - S_{i(j+1)}} & (j = 2, S_{ij} \leq u_i \leq S_{i(j+1)}) \\ 3 + 3 \times \frac{S_{ij} - u_i}{S_{ij} - S_{i(j+1)}} & (j = 3, S_{ij} \leq u_i \leq S_{i(j+1)}) \\ 6 + 4 \times \frac{S_{ij} - u_i}{S_{ij} - S_{i(j+1)}} & (j = 4, S_{ij} \leq u_i \leq S_{i(j+1)}) \\ 10 & (j = 4, S_{ij} \leq u_i \leq S_{i(j+1)}) \end{cases} \tag{1}$$

Equation (1): $u_i$ is the measured value of the single component I of the water sample u; $S_{ij}$ is the standard level j for individual component i of the water sample. As the quality standards IV and V have the same values, the artificial rule is that $S_{i5} = 2S_{i4}$.

② Modification of $F_{ave}$. The NEMEROW index evaluation method particularly highlights the impact of the extreme values, i.e., water quality indicators with more serious exceedances, on the environmental quality of groundwater while ignoring the role of the average score. To avoid the impact of the above situation, it can be modified for $F_{ave}$ as follows.

$$F_{ave} = \frac{1}{m} \sum_1^m F_i \tag{2}$$

Equation (2): $F_{ave}$ is the Mean of the first m items of the rating value; m is determined based on practical experience; when $n \leq 5$, $m = 5$; when $n > 5$, m is generally taken as 5. In areas with severe groundwater pollution, the value of m is appropriately increased, depending on the actual situation.

## 3. Results and Discussion

*3.1. Water Chemistry Characterization*

The results of the statistical analysis of the sample test data were shown in Table 1. The box plots are used to better describe and visualize these characteristics and variations (Figure 3) [37]. Its values represent mainly the top quartile, median and bottom quartile. Its values from top to bottom are mainly the upper quartile, median and lower quartile. The

results show that the mean pH lies between 7.0 and 8.0, indicating that the groundwater is neutral to weakly alkaline.

**Table 1.** Statistics of hydrochemical parameters of groundwater (unit: mg/L, except for pH).

| Type | Parameter | pH | TDS | TH | $Ca^{2+}$ | $Mg^{2+}$ | $K^+$ | $Na^+$ | $HCO_3^-$ | $SO_4^{2-}$ | $Cl^-$ | $NO_3^-$ |
|------|-----------|-----|-----|-----|------|------|-----|------|------|------|------|------|
| Fracture water ($n = 19$) | Mean | 7.34 | 998.81 | 380.6 | 94.23 | 34.71 | 1.78 | 199.04 | 104.33 | 135.62 | 385.94 | 35.94 |
| | SD | 0.27 | 2032.16 | 670.59 | 200.68 | 46.61 | 2.49 | 512.01 | 82.42 | 218.81 | 1103.45 | 32.57 |
| | Min | 6.9 | 67.21 | 46.12 | 12.93 | 3.36 | 0.13 | 2.17 | 7.65 | 3.05 | 21.53 | 0 |
| | Max | 8.2 | 8877.64 | 3064.85 | 910.97 | 191.9 | 8.82 | 2125 | 306.08 | 778.46 | 4812.71 | 101.35 |
| Pore groundwater ($n = 29$) | Mean | 7.2 | 1435.83 | 554.26 | 88.35 | 81.04 | 12.92 | 277.95 | 108.77 | 235.23 | 560.83 | 69.69 |
| | SD | 0.38 | 3211.69 | 802.72 | 63.12 | 163.27 | 36.36 | 927.56 | 75.61 | 403.37 | 1722.63 | 67.98 |
| | Min | 6.5 | 74.83 | 39.2 | 8.31 | 4.48 | 0.17 | 6.67 | 15.3 | 10.68 | 13.13 | 0.36 |
| | Max | 8.5 | 17,138.1 | 3597.12 | 263.18 | 728.01 | 200 | 5000 | 369.84 | 1617.98 | 9160.7 | 244.75 |
| Surface water ($n = 5$) | Mean | 7.3 | 301.99 | 138.71 | 40.88 | 8.9 | 4.43 | 30.69 | 85.07 | 56.11 | 40.91 | 23.25 |
| | SD | 0.32 | 302.26 | 127.06 | 35.8 | 9.18 | 5.75 | 41.82 | 90.26 | 62.81 | 53.38 | 33.27 |
| | Min | 6.8 | 91.69 | 48.54 | 15.43 | 2.31 | 0.16 | 4.43 | 27.98 | 19.21 | 7.18 | 5.52 |
| | Max | 7.6 | 814.84 | 351.78 | 100.8 | 24.31 | 12.59 | 103.6 | 243.46 | 165.7 | 132.78 | 82.6 |

Note: Min is the minimum value, Max is the maximum value, Mean is the Mean, and SD is the standard deviation.

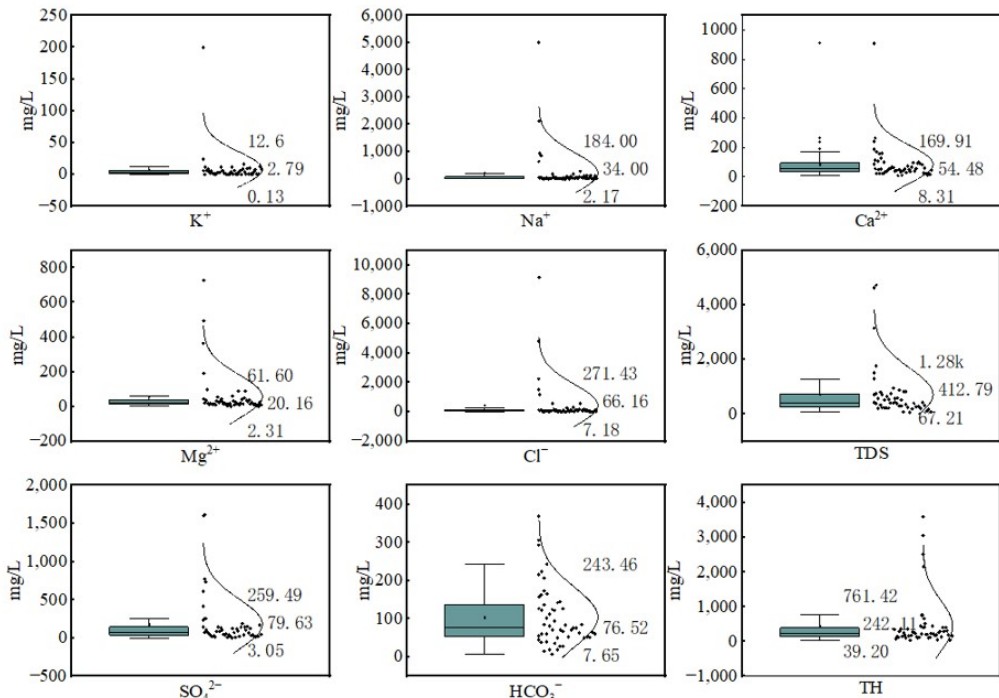

**Figure 3.** The box diagram of the hydrochemical composition of groundwater.

The $K^+$ mass concentration ranged from 0.13 to 200 mg/L, with a mean value of 0.92 mg/L and the majority below 12.6 mg/L; The $Na^+$ mass concentration ranged from 2.17~5000 mg/L, with a mean value of 212 mg/L and the majority below 184 mg/L, only a few points with high concentrations (LS029) indicating that the relative stability of $Na^+$ in groundwater is relatively bad and varies widely over the region.

In the fractured water, $Ca^{2+}$ concentrations ranged from 12.93 to 910.97 mg/L; compared to concentrations from 8.31 to 263.18 mg/L in the pore water, indicating that $Ca^{2+}$ is relatively stable in the pore water of the Quaternary, with little regional variation.

The $Mg^{2+}$ mass concentration ranged from 3.36 to 728 mg/L, with the majority below 169.91 mg/L. The ranges of $Cl^-$ mass concentration are comparatively large, from 13.13

to 9160.7 mg/L, with the majority below 271.43 mg/L. However, there were 10 samples (LS001, LS003, LS010, LS026, LS029, LS035, LS071, LS077, LS082, and LS090) whose mass concentrations exceeded the limit value (250 mg/L) of the Groundwater Quality Standard (GB/T 14848-2017) [33] and the cut-off value (250 mg/L) for the presence or absence of seawater intrusion, indicate that the groundwater may be affected by a certain degree of seawater intrusion. The $Cl^-$ concentration in fracture water is much lower than that in pore water, which means that rainfall has reduced the concentration of $Cl^-$ in groundwater.

The $SO_4^{2-}$ mass concentration was maximized to 1602.71 mg/L, with a mean value of 194 mg/L, and mostly concentrated in the range of 3.36 to 259.49 mg/L, indicating that $SO_4^{2-}$ is less stable both in fracture water and in pore water.

The distribution range of TDS was relatively large, with a maximum value of 17,138.1 mg/L. According to the classification of TDS [38], the groundwater exists as freshwater, brackish water, saline water, and in some areas, saline.

### 3.2. Water Chemistry Types

The Shukarev classification [39] is based on the major ions in groundwater $Ca^{2+}$, $Mg^{2+}$, $Na^+$ ($K^+$ combined in $Na^+$), $HCO_3^-$, $Cl^-$, $SO_4^{2-}$ and TDS, which are classified according to the combination of anions and cations with milligram equivalent percentages greater than 25%, and are divided into 49 types of water (Table 2).

**Table 2.** Schukarev's classification of type 49 water.

| More than 25% Milligram Equivalent Ion | HCO₃ | HCO₃ + SO₄ | HCO₃ + SO₄ + Cl | HCO₃ + Cl | SO₄ | SO₄ + Cl | Cl |
|---|---|---|---|---|---|---|---|
| Ca | 1 | 8 | 15 | 22 | 29 | 36 | 43 |
| Ca + Mg | 2 | 9 | 16 | 23 | 30 | 37 | 44 |
| Mg | 3 | 10 | 17 | 24 | 31 | 38 | 45 |
| Na + Ca | 4 | 11 | 18 | 25 | 32 | 39 | 46 |
| Na + Ca + Mg | 5 | 12 | 19 | 26 | 33 | 40 | 47 |
| Na + Mg | 6 | 13 | 20 | 27 | 34 | 41 | 48 |
| Na | 7 | 14 | 21 | 28 | 35 | 42 | 49 |

The hydrochemical classification of groundwater in the study area using the Shukarev classification was $HCO_3$-Ca·Mg, $HCO_3$-Na·Ca, $HCO_3$·$SO_4$-Ca, $HCO_3$·$SO_4$-Ca·Mg, $HCO_3$·$SO_4$·Cl-Ca·Mg, $HCO_3$·$SO_4$·Cl-Na·Ca·Mg, $HCO_3$·Cl-Ca, $HCO_3$·Cl-Ca·Mg, $HCO_3$·Cl-Na·Ca·Mg, $SO_4$-Ca·Mg, $SO_4$·Cl-Mg, $SO_4$ ·Cl-Ca·Mg, $SO_4$ ·Cl-Na·Ca·Mg, Cl-Ca, Cl-Ca·Mg, Cl-Na·Ca·Mg, and Cl-Na·Ca. The chemical type share is shown in Figure 4. Among them, Cl type accounted for 29%, $SO_4$·Cl type for 21%, $HCO_3$·Cl type for 25%, and $HCO_3$·$SO_4$·Cl type for 9%.

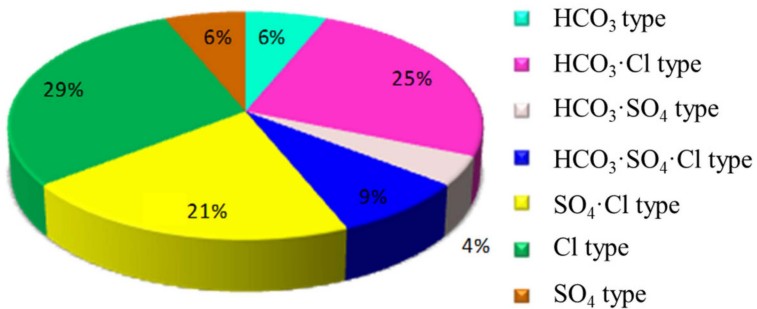

**Figure 4.** The pie chart of groundwater chemistry types.

The Piper trilinear diagram [40] can reflect the changes in the main ionic composition and water chemistry characteristics of groundwater and can be used to classify the water

chemistry types. The results of the dissolved ion analysis of the groundwater samples are shown in the Piper diagram in Figure 5. From the Piper trilinear diagram, the cations are mainly located in 3 regions, A, B, and D, and the anions are primarily located in areas B and G. Incorporating the Shukarev water chemistry classification, the water chemistry types are majorly Cl·SO$_4$-Na and SO$_4$·Cl-Ca·Mg.

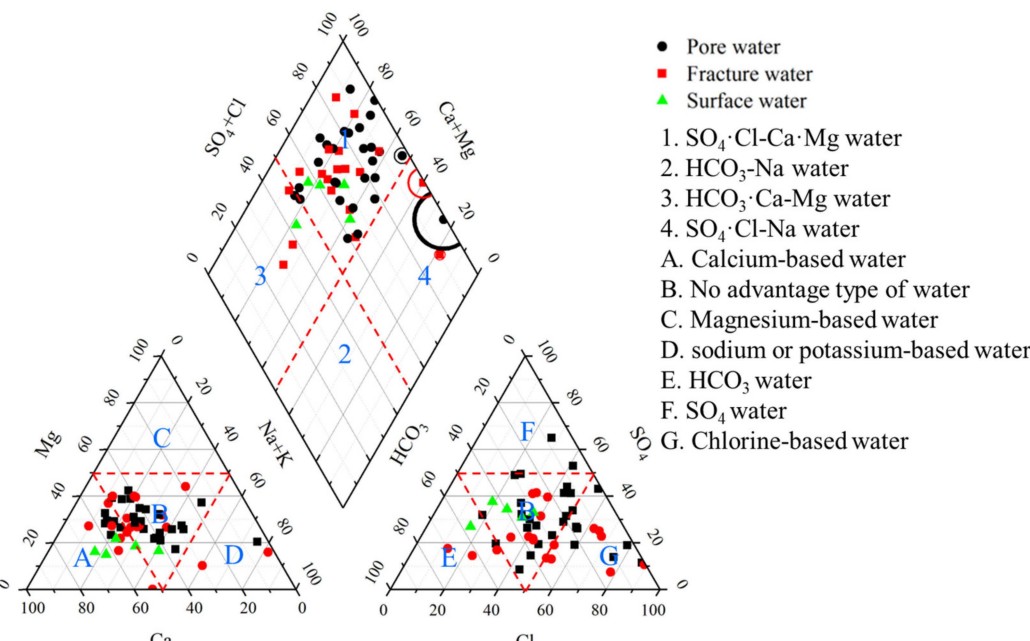

**Figure 5.** Piper diagram of groundwater hydrochemistry.

Another useful parameter for classifying water quality is the Ionic Salinity or Total Ionic Salinity (TIS), representing the sum of the concentrations of the major anions and cations expressed in meq/L [41]. Figure 6 shows most samples had TIS below 80, and only a few showed high salinity. LS029 and LS033 showed very high Cl/SO$_4$ ratios, giving more reason to assume that seawater intrusion is occurring in some areas.

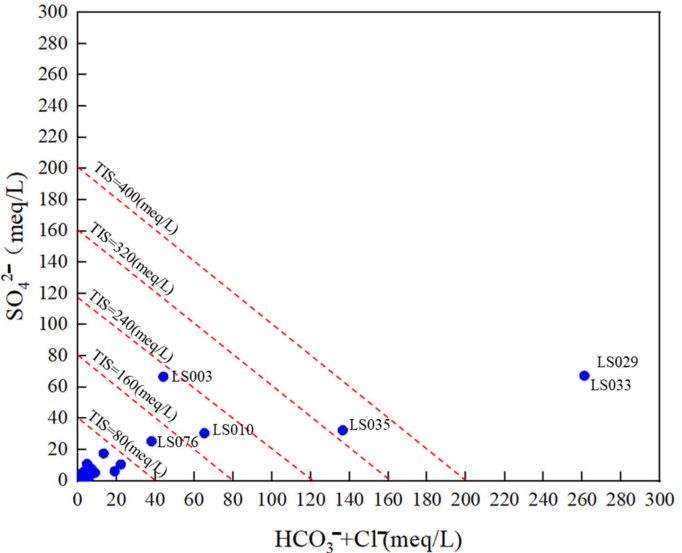

**Figure 6.** Correlation diagram of SO$_4$ vs. HCO$_3$ + Cl for the groundwater samples. Isosalinity lines are drawn for reference.

As shown in Figure 7, the $HCO_3^-$ type groundwater is mainly located in the western part of the study area, i.e., in the Laoshan Mountains and the upper reaches of the river. The lithology at these locations is mainly granite and gneiss. Controlled by topography, it has rapid groundwater runoff rates. The anions in groundwater are mainly $HCO_3^-$ for, and the cations are mainly $Ca^{2+}$ and $Na^+$, forming $HCO_3$-Ca·Na type freshwater and, where localized, $HCO_3$-Ca·Mg type freshwater.

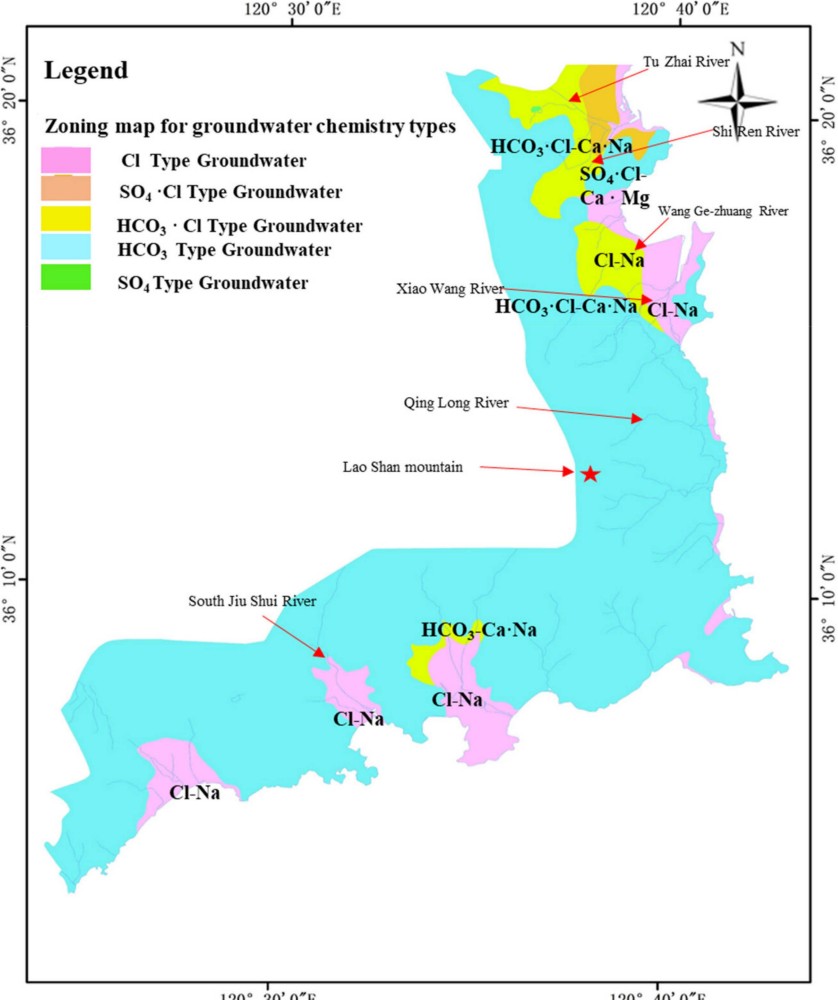

**Figure 7.** The zoning map of groundwater chemistry types.

The $HCO_3$·Cl type groundwater is mainly distributed in the Tuzhai River basin, Wanggezhuang River, and the middle and upper reaches of Dahedong-Xiaohedong, and is the main hydrochemical type of shallow freshwater. The anions are mostly $HCO_3^-$ and $Cl^-$, and the cations are mainly $Ca^{2+}$ and $Na^+$, which forms $HCO_3$·Cl-Ca·Na type groundwater. It forms $HCO_3$·Cl-Ca types of freshwater in places with good water quality.

Chloride-type groundwater is found near the entrance to the river and in the eastern part of Wanggezhuang. The area is adjacent to the Huanghai, and the $Cl^-$ is dominated by the chlorinated salts of seawater, and the cations are mainly $Ca^{2+}$ and $Na^+$, forming Cl-Ca·Na-type groundwater. In the offshore GangDong, the groundwater is Cl-Na type. The mineralization is highly variable, and TDS is mostly less than 1000 mg/L, locally up to 17,000 mg/L. Under the influence of agriculture, nitrate exceeds the standard, with a maximum of 187.1 mg/L.

*3.3. Mechanisms of Groundwater Hydrochemistry*

3.3.1. Ionic Correlation Analysis

Correlation analysis provides an analysis of multiple sets of variable elements, which reflects the degree of correlation between ions, and the $|r|$ closer to 1, the significance is stronger, and the correlation is greater [42]. Table 3 shows the results of the correlation analysis of the chemical fractions of groundwater in the study area. The magnitude of the correlation between TDS and cations is $Na^+ > Mg^{2+} > K^+ > Ca^{2+}$, and the correlation coefficients are 0.987, 0.901, 0.864, and 0.612, respectively, while with anions were $Cl^- > SO_4^{2-}$ and the correlation coefficients were 0.994 and 0.860, respectively. In particular, TDS correlated very strongly with TH, $Na^+$, $Mg^{2+}$, $K^+$, $Ca^{2+}$, $Cl^-$, and $SO_4^{2-}$, and very weakly with $HCO_3^-$ and $NO_3^-$, indicating that the soluble components are the main determinants of TDS.

**Table 3.** The correlation coefficient of the chemical composition of groundwater.

| | pH | TDS | TH | $Ca^{2+}$ | $Mg^{2+}$ | $K^+$ | $Na^+$ | $HCO_3^-$ | $SO_4^{2-}$ | $Cl^-$ | $NO_3^-$ |
|---|---|---|---|---|---|---|---|---|---|---|---|
| PH | 1 | | | | | | | | | | |
| TDS | 0.000 | 1.000 | | | | | | | | | |
| TH | −0.075 | 0.924 ** | 1.000 | | | | | | | | |
| $Ca^{2+}$ | −0.108 | 0.612 ** | 0.765 ** | 1.000 | | | | | | | |
| $Mg^{2+}$ | −0.037 | 0.901 ** | 0.918 ** | 0.447 ** | 1.000 | | | | | | |
| $K^+$ | 0.048 | 0.864 ** | 0.668 ** | 0.202 | 0.804 ** | 1.000 | | | | | |
| $Na^+$ | 0.031 | 0.987 ** | 0.853 ** | 0.538 ** | 0.852 ** | 0.902 ** | 1.000 | | | | |
| $HCO_3^-$ | 0.354 * | 0.171 | 0.132 | 0.065 | 0.144 | 0.159 | 0.162 | 1.000 | | | |
| $SO_4^{2-}$ | −0.069 | 0.860 ** | 0.921 ** | 0.525 ** | 0.956 ** | 0.687 ** | 0.792 ** | 0.198 | 1.000 | | |
| $Cl^-$ | 0.010 | 0.994 ** | 0.898 ** | 0.619 ** | 0.867 ** | 0.866 ** | 0.992 ** | 0.125 | 0.806 ** | 1.000 | |
| $NO_3^-$ | −0.477 ** | −0.127 | −0.079 | −0.056 | −0.073 | −0.026 | −0.156 | −0.266 | −0.031 | −0.160 | 1 |

Note: * indicates a significant correlation at the 0.05 level; ** indicates a significant correlation at the 0.01 level.

TH also showed an extremely strong correlation with $Na^+$, $Mg^{2+}$, $K^+$, $Ca^{2+}$, $Cl^-$, and $SO_4^{2-}$, which indicates that there is a common source that might have occurred with the dissolution of rock salt, $CaSO_4$, and $CaSO_4 \cdot 2H_2O$. Of these, the $Na^+$ correlation with $Cl^-$ reached 0.992, which may be caused by seawater intrusion.

3.3.2. Gibbs Diagrammatic Model

The Gibbs plot [43] can qualitatively reflect the controlling factors of groundwater ion characteristics from a macroscopic perspective. The Gibbs diagram classifies the control types of groundwater hydrochemical components as precipitation-controlled, rock-weathering, and evaporation-concentration types [44]. From Figure 8a,b, it can be seen that the majority of groundwater sampling points are mainly located in the rock weathering type area, and some points fall in the evaporation-crystallization zone, far from the atmospheric precipitation zone, indicating that the water chemical ions in the area are less influenced by atmospheric precipitation effects, while water-rock interaction and evaporation-concentration are the controlling factors of water chemistry ions in the study area.

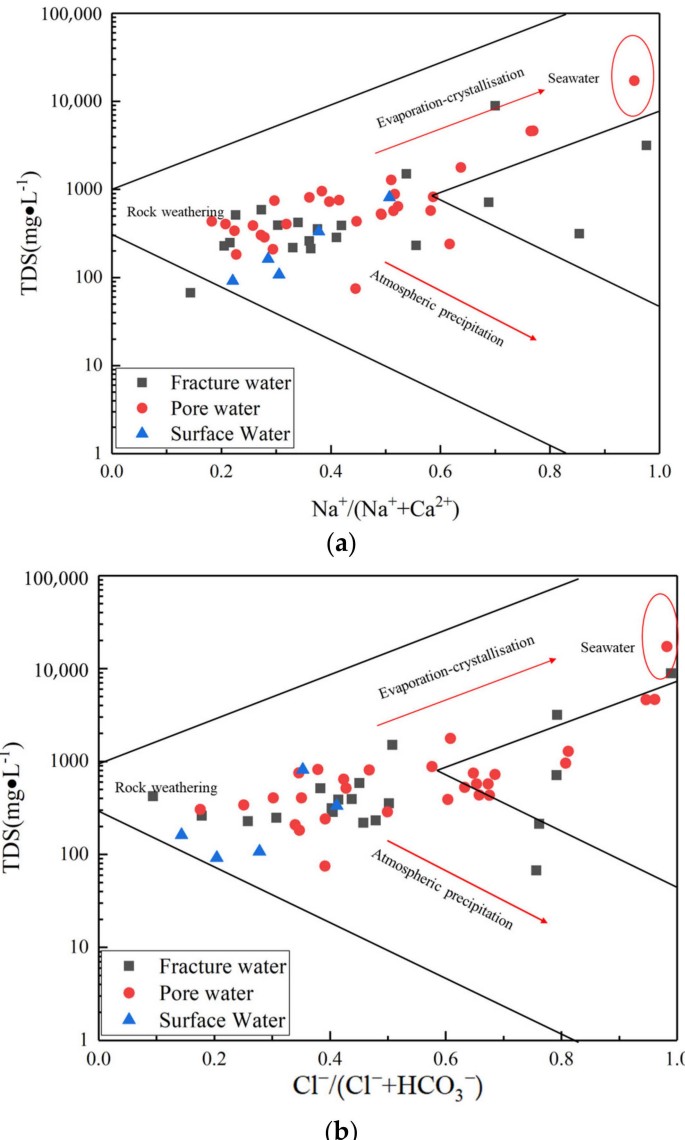

**Figure 8.** Gibbs diagram of groundwater. (**a**) The relationship between the TDS and $Na^+/(Na^+ + Ca^{2+})$. (**b**) The relationship between the TDS and $Cl^-/(Cl^- + HCO_3^-)$.

### 3.3.3. Analysis of Groundwater Mineral Phase Balance

Depending on the SI value, we can determine the reaction state between water and rocks and minerals [45].

$$SI = \lg \frac{IAP}{K} \tag{3}$$

where IAP is the activity product of the relevant ions in the mineral dissolution reaction; K is the equilibrium constant of the reaction. SI > 0 means the mineral is in a super-saturated state relative to the aqueous solution, indicating that the mineral will precipitate out of the solution; SI < 0, the mineral is reactive and can continue to dissolve in the solution until it reaches the equilibrium state; SI = 0 means the aqueous solution and the mineral are exactly in the equilibrium state [45]. Generally, we consider $-0.5 < SI < +0.5$ as the equilibrium state, less than $-0.5$ as the dissolved state, and more than 0.5 as the saturated state.

To obtain the precipitation-dissolution status of groundwater minerals in the study area, we calculate the saturation index (SI) of anhydrite, aragonite, calcite, dolomite, gypsum, and halite minerals in groundwater with the help of PHREEQC. We obtained the saturation indices of six minerals, namely $CaSO_4$, $CaCO_3$, $CaCO_3$, $CaMg(CO_3)_2$, $CaSO_4:2H_2O$, and NaCl, and plotted the box plots separately, as shown in Figure 9.

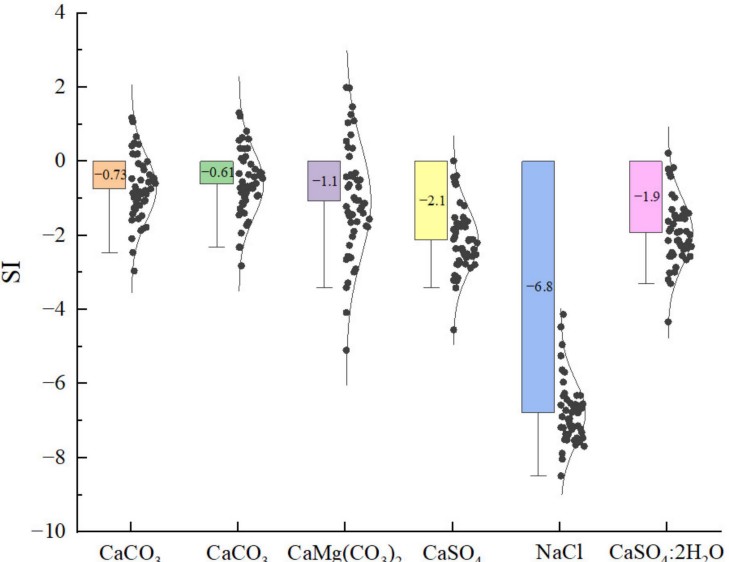

**Figure 9.** The box diagram of the Mineral saturation index.

The SI of calcite (CaCO₃) and dolomite (CaMg(CO₃)₂) are all less than 0, except LS010, LS016, LS026, LS029, LS033, LS035, LS073, LS077, LS082, LS090, LS104, and LS110, which are greater than 0. It indicates that calcite and dolomite in the study area are partially in the supersaturated state and partially in the dissolved state. The SI of calcite and dolomite varied from −2.82 to 1.31 and from −5.1 to 1.99, with mean values of −0.6 and −1.07. Combining the sampling distribution map and the trend of the SI index (Figure 10), it can be seen that (The locations of all sampling sites are arranged from inland to coastal), the SI of calcite and dolomite along the direction of groundwater seepage is gradually increasing, indicating that calcite and dolomite gradually change to the supersaturated state along the seepage direction.

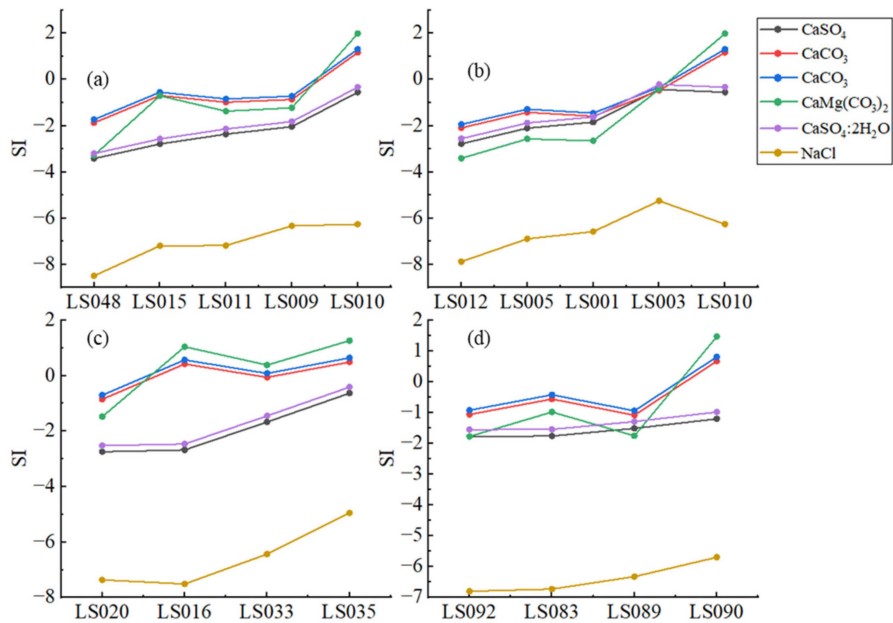

**Figure 10.** The trend graph of the SI index from inland to coast. (**a**) The trend graph of the SI index from LS048 to LS010. (Shi Ren River). (**b**) The trend graph of the SI index from LS012 to LS010. (Tu Zhai River). (**c**) The trend graph of the SI index from LS020 to LS035. (Wang Ge Zhuang River). (**d**) The trend graph of the SI index from LS092 to LS090. (Da He Dong River).

The SI of gypsum (CaSO$_4$:2H$_2$O) is less than 0 except LS029, which indicates that the gypsum is still in the dissolved state and has not reached the saturation state in the study area. The saturation index of gypsum varied from −4.55 to 0.01, with a mean value of −2.12. Combining with the sampling distribution, it can be seen that the SI of gypsum showed an increasing trend along the seepage path, indicating that the gypsum was gradually changing from the dissolved state to the equilibrium state.

The SI of halite(NaCl) is less than 0, which means that the Halite(NaCl) is still in the dissolved state in the study area and has not reached the saturation state. The range of saturation index is −4.13~−8.49, with an average value of −6.7, where the dissolution is the most intense. Combined with the sampling distribution, it can be seen that the SI of Halite(NaCl) shows an increasing trend along the seepage path, indicating that the gypsum is gradually in the dissolved state to the equilibrium state.

The correlations between the SI of calcite (CaCO$_3$), dolomite (CaMg(CO$_3$)$_2$), halite (NaCl), and gypsum (CaSO$_4$:2H$_2$O) are shown in Figure 11. The saturation indices among calcite and dolomite, and rock salt and gypsum have a good linear correlation (Figure 11a,d, indicating that the dissolution of calcite and dolomite, halite and gypsum by groundwater may be synchronous and belong to the synchronous dissolution effect.

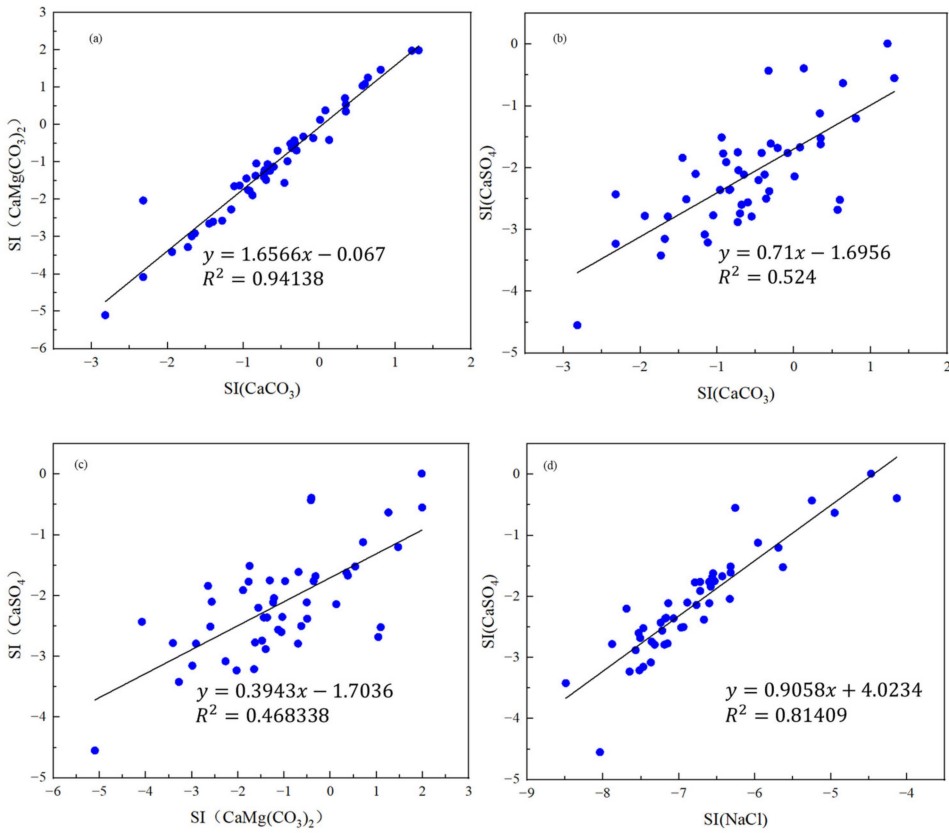

**Figure 11.** *Cont.*

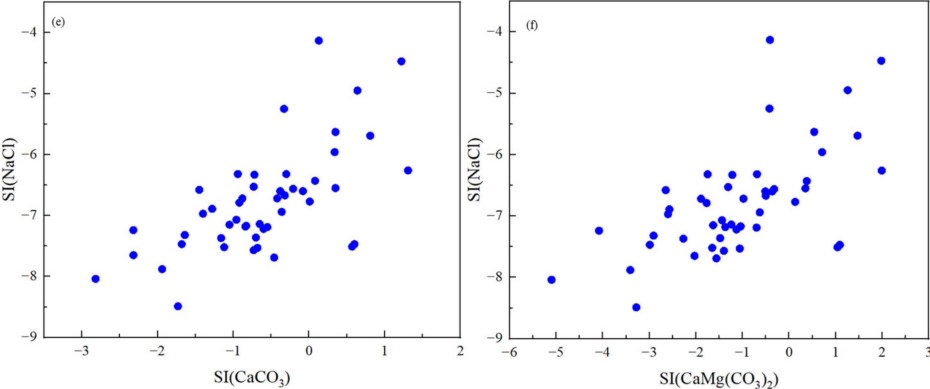

**Figure 11.** The graph of the relationship between the saturation index of each mineral. (**a**) Ratio diagram of saturation indices for $CaCO_3$ and $CaMg(CO_3)_2$. (**b**) Ratio diagram of saturation indices for $CaCO_3$ and $CaSO_4$. (**c**) Ratio diagram of saturation indices for $CaMg(CO_3)_2$ and $CaSO_4$. (**d**) Ratio diagram of saturation indices for NaCl and $CaSO_4$. (**e**) Ratio diagram of saturation indices for $CaCO_3$ and NaCl. (**f**) Ratio diagram of saturation indices for $CaMg(CO_3)_2$ and NaCl.

The SI of calcite and gypsum has a certain correlation (Figure 11b), and the larger the SI of calcite, the higher the SI of gypsum, indicating that the dissolution of both may be established through an intermediate process (e.g., de-dolomitization). The SI of dolomite has no obvious direct correlation with the gypsum, calcite ($CaCO_3$), dolomite ($CaMg(CO_3)_2$), and rock salt saturation index (Figure 11c,e,f).

### 3.3.4. Ion Ratio Analysis

The ion ratios between different chemical species have frequently been used to evaluate seawater intrusion in coastal areas [46]. The ratios of $\gamma$ ($Na^+/Cl^-$), $\gamma$ ($Cl^-/HCO_3^-$), and $\gamma$ ($Cl^-/SO_4^{2-}$) in groundwater concerning $Cl^-$ are shown in Figure 12. The $\gamma$ ($Na^+/Cl^-$) ranged from 0.1 to 1.7, with the mean value of $\gamma$ ($Na^+/Cl^-$) in the pore water being 0.72, indicating the presence of seawater intrusion. The anions in seawater are mainly $Cl^-$, while inland, they are $HCO_3^-$ and $SO_4^{2-}$, therefore the analysis of $\gamma$ ($Cl^-/HCO_3^-$) and $\gamma$ ($Cl^-/SO_4^{2-}$) gives a better indication of the extent of seawater intrusion in the region. The $\gamma$ ($Cl^-/HCO_3^-$) ratios showed that five samples had ratios < 0.5 [47], indicating no seawater intrusion; 39 samples had ratios between 0.36 and 6.5, indicating slight intrusion; and the remaining nine samples had ratios > 6.6 [48], indicating substantial seawater intrusion. The three strongly intruded water samples were all Quaternary pore water distributed in coastal areas.

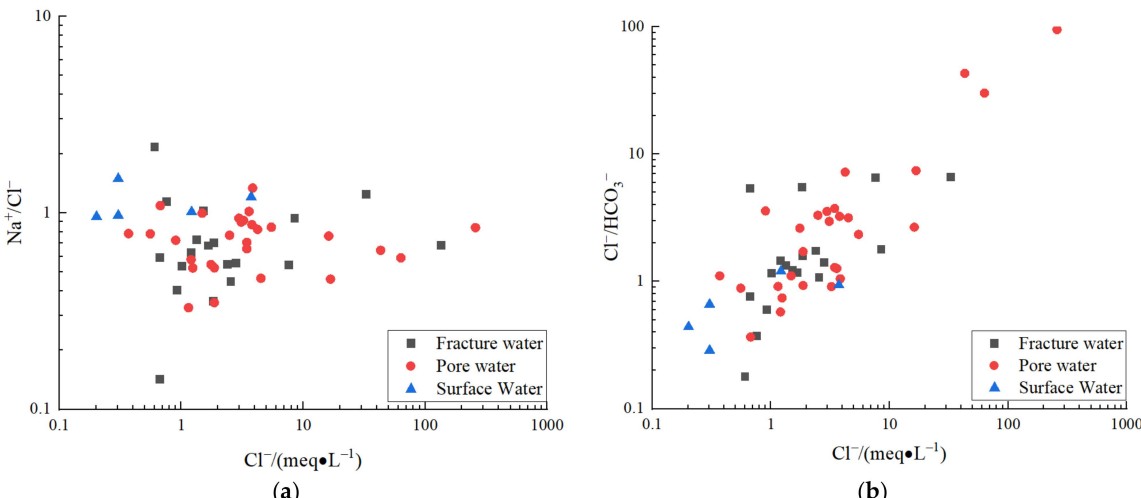

**Figure 12.** *Cont.*

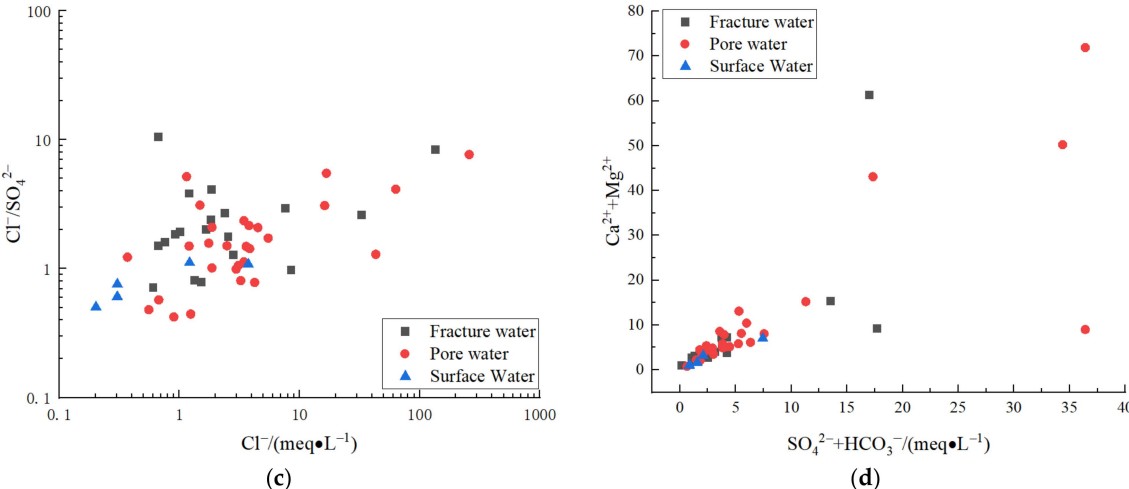

**Figure 12.** Relation diagrams of significant ion ratio in groundwater. (**a**) Scatter plot of $Cl^-$ vs. $Na^+/Cl^-$. (**b**) Scatter plot of $Cl^-$ vs. $Cl^-/HCO_3^-$. (**c**) Scatter plot of $Cl^-$ vs. $Cl^-/SO_4^{2-}$. (**d**) Scatter plot of $(SO_4^{2-} + HCO_3^-)/(Ca^{2+} + Mg^{2+})$.

The relationship between $Ca^{2+} + Mg^{2+}$ and $SO_4^{2-} + HCO_3^-$ is shown in Figure 12c. Most of the sampling Points are located above or near the Y = X line. Nearly 90% of the sampling Points $(Ca^{2+} + Mg^{2+})/(SO_4^{2-} + HCO_3^-)$ ratio coefficient is greater than 1, suggesting that $Ca^{2+}$ and $Mg^{2+}$ in groundwater mainly originated from the dissolution of carbonate rocks. The sampling sites with $(Ca^{2+} + Mg^{2+})/(SO_4^{2-} + HCO_3^-)$ ratio coefficients less than 1 accounted for 10% of the total, indicating that groundwater needs to be kept in ionic balance by the dissolution of silicate and evaporite.

### 3.4. Water Quality Evaluation Results

According to the Groundwater Quality Standard (GB/T 14848-2017) [35], this paper has six items such as $Na^+$, $SO_4^{2-}$, $NO_3^-$, $Cl^-$, TDS, and TH were selected as groundwater quality evaluation factors. Groundwater quality was classified into five categories using the modified NEMEROW index method of a comprehensive evaluation. Table 4 shows the water quality evaluation results.

**Table 4.** Summary of groundwater quality assessment results.

| Number | One-Factor Index Evaluation | Modified NEMEROW Index Method | Number | One-Factor Index Evaluation | Modified NEMEROW Index Method | Number | One-Factor Index Evaluation | Modified NEMEROW Index Method |
|---|---|---|---|---|---|---|---|---|
| LS001 | V | V | LS030 | IV | I | LS077 | V | II |
| LS003 | V | IV | LS033 | V | II | LS081 | III | I |
| LS005 | V | III | LS035 | V | IV | LS082 | V | V |
| LS009 | V | IV | LS040 | V | IV | LS083 | V | II |
| LS010 | V | III | LS041 | V | IV | LS084 | IV | II |
| LS011 | V | III | LS043 | V | II | LS085 | IV | II |
| LS012 | V | II | LS046 | V | II | LS089 | V | V |
| LS015 | V | II | LS048 | III | I | LS090 | V | II |
| LS016 | II | I | LS051 | V | II | LS092 | V | III |
| LS020 | III | I | LS060 | V | III | LS096 | V | III |
| LS022 | III | I | LS063 | I | I | LS097 | V | IV |
| LS023 | V | II | LS071 | V | II | LS102 | V | II |
| LS024 | IV | II | LS073 | III | I | LS-104 | V | V |
| LS026 | V | II | LS074 | III | I | LS-110 | III | I |
| LS027 | V | IV | LS075 | V | II | LS-111 | III | I |
| LS029 | V | V | LS076 | V | II | LS-19 | II | I |

The results showed that the groundwater quality in the study area was mainly III, IV, and V water according to the single factor index evaluation method.

Using the NEMEROW formula for the evaluation of groundwater quality in the study area showed that groundwater category I–V water existed in the study area, and each type of water accounted for 25%, 37.5%12.5%, 14.58%, and 10.42% of the overall samples monitored, respectively. The overall water quality was good, showing that the modified NEMEROW index method can objectively evaluate groundwater quality.

Based on kriging interpolation, we used ArcGIS 10.1 to map the groundwater quality zoning (Figure 13). As can be seen from Figure 13, the Class V water is mainly pore water, which is distributed Downstream of the Xiaowang River-Gangdong—Houtong area, the South of Dengying-KaoLao Island-the downstream of Liangshui River, and the area around Shazikou Town in the lower reaches of Nanjiushui River. It is distributed along the coastline in the form of a belt.

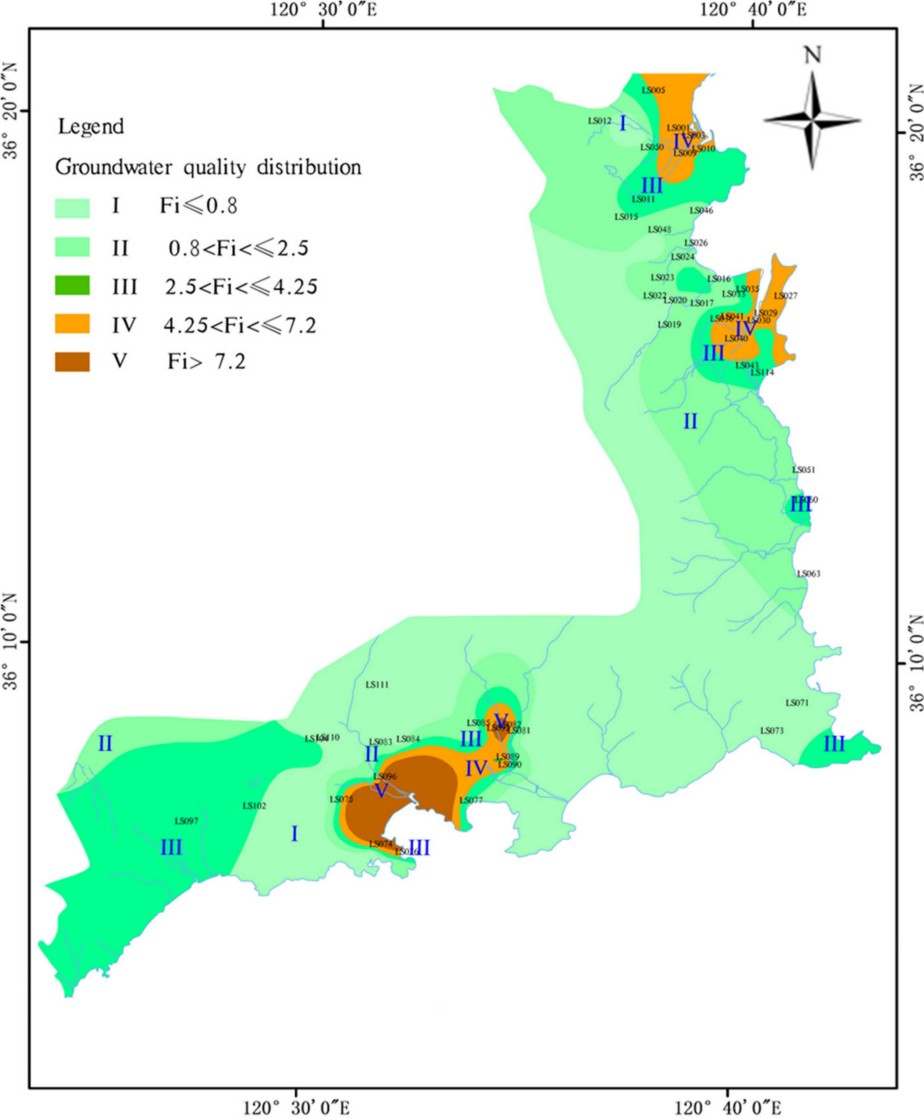

**Figure 13.** Groundwater quality distribution.

The seawater intrusion has led to increased $Cl^-$ content in groundwater. Combined with the correlation analysis results, $Ca^{2+}$, $SO_4^{2-}$ and $HCO_3^-$ concentrations in groundwater may also be affected by seawater-groundwater mixing, leading to an increase in content and worsening groundwater quality. The IV is mainly located in the downstream regions of the Tuzhai River, Shiren River, and Xiaowang River. These are populated and

significant agricultural areas where human activities and massive application of chemical fertilizers and pesticides are the main factors causing groundwater pollution. The main components in groundwater that exceed the standard are TDS, $Cl^-$, $NO_3$-, and $SO_4^{2-}$, etc. Thus, seawater intrusion and anthropogenic pollution are the leading causes of water quality deterioration in the area.

To conclude, the groundwater quality in the coastal area of Laoshan District is generally good, with 70.76% of the groundwater area above the excellent level. The water quality changes gradually from inland to the coastline, especially in the towns and villages at the mouth of the rivers, which are affected by agricultural and domestic sewage and seawater intrusion, and the groundwater quality is poor.

## 4. Conclusions

This paper takes the 2016–2017 groundwater quality survey data as the background and uses the concentration of major ions in groundwater as the water chemistry index to analyze the chemical characteristics and evolution pattern of groundwater in Laoshan District, Qingdao City, and obtain the following conclusions.

(1) The groundwater is predominantly weakly alkaline. $Na^+$ is relatively unstable in groundwater and varies widely regionally. $Ca^{2+}$ is less stable in bedrock fracture water and relatively stable in Quaternary pore water. $HCO_3^-$ is relatively stable in groundwater. The $Cl^-$ mass concentration ranges over a wide range, and the mean value exceeds the cut-off value for the presence or absence of seawater intrusion (250 mg/L), indicating that there is a certain seawater intrusion and that the $Cl^-$ concentration in the bedrock fracture water is much lower than the pore water, suggesting that rainfall has reduced the groundwater $Cl^-$ concentration. According to the classification of TDS, the groundwater exists as freshwater, brackish water, saline water, and in some areas, saline.

(2) According to the Piper diagram, the groundwater chemistry type is dominated by the Cl·SO4-Na and SO4·Cl-Ca·Mg types. Along the runoff path, the groundwater chemistry type (from west to east) changes from $HCO_3$-Ca·Na to Cl-Ca·Na. TDS was strongly correlated with TDS, $Na^+$, $Mg^{2+}$, $K^+$, $Ca^{2+}$, $Cl^-$, and $SO_4^{2-}$, and the saturation index showed a gradual increase along the groundwater drainage flow path, it indicates that the main groundwater ions originate from the dissolution of halite, sulfate, and carbonate. Combining the Gibbs diagram, mineral saturation index, and ion relationship diagram to analyze the sources of major groundwater ions, it was determined that the main factor controlling the chemical composition of groundwater in the area is rock weathering, in addition to seawater intrusion and human activities are important factors affecting the groundwater environment in Laoshan District, Qingdao.

(3) Groundwater quality is mainly Class III water, with good quality overall. Class IV and V water are mainly distributed in the middle and lower reaches of the rivers, in a band along the coastline. The results show that it is necessary to further strengthen the monitoring of groundwater in the study area. We should continue to consolidate the prevention and control of seawater intrusion in coastal areas and ensure the ecological balance of the groundwater environment.

**Author Contributions:** Methodology, T.J., Y.D., L.X. and S.G. (Shuai Gao); Writing—original draft, S.G. (Shenting Gang). All authors have read and agreed to the published version of the manuscript.

**Funding:** This research was funded by the National Natural Science Foundation of China (42272288).

**Data Availability Statement:** Data will be made available on request.

**Conflicts of Interest:** The authors declare no conflict of interest.

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
