# Peer review of "Hydrochemical Characteristics and Formation Mechanism of Groundwater in Qingdao City, Shandong Province, China"

_water, doi:10.3390/w15071348_

Round 1

Author Response

Dear Editor and Reviewers,

We sincerely appreciate all of you for the time and effort that you have put into reviewing our manuscript entitled " Hydrochemical characteristics and formation mechanism of groundwater in Qingdao City, Shandong Province, China" (water-2285381). Your comments and suggestions are very important and helpful in improving our work. We have revised our manuscript carefully according to these comments and all of the revisions are highlighted in red in the “revised manuscript”. A point-by-point response to the comments and suggestions made by the reviewers is attached.

Reviewer 2 Report

The purposes of the manuscript water-2285381  are to assess the hydrochemical characteristics of groundwater in in Laoshan District, Qingdao through Piper's trilinear diagram, Gibbs' diagram, correlation analysis, ion ratio, and modified Nemerov index method

 The paper appears no well-structured and some sections must be improved.  Therefore, I believe 

the manuscript should be published only after major revision.

 Comments (R = row#):

R=25: The introduction main title is absent

R=38: The authors say: Currently, the main research methods related to groundwater chemical characteristics and their influencing factors are mathematical statistics, graphical procedures, ion ratio method, and mineral saturation index method...but not only ….. the geochemical modelling of water-rock interaction processes which represents one of the most powerful tools. Authors look for example at the works of: [1], [2]

R=72: Geogenic pollution it is little elaborated in the introduction. I recommend reading the following works and integrating this part: [3]

R=87: there is no geological framework with a petrographic and mineralogical description of the main lithologies outcropped. A geological map, also schematic of the area, should be inserted!

R=116: add precision and analytical accuracy data if present

R=118: in figure 2 the geology can also be inserted

R=121: the authors write: I plotted…., but the paper wasn't written by a single author, so you should write: we plotted…..

R=156: this part should be explained better, referring to the original literature works!

R=169: use box plots to better describe and visualize these characteristics and variations

R=187: To evaluate the chemical composition of the water it is not enough to use the Piper diagram because it does not take into account (as proposed by the authors) salinity, I suggest using a TIS salinity diagram, as proposed by: [4]

R=192: figure 4 not 5

R=198: indicate the salinity of these waters by referring to the TIS diagram

R=201: from figure 4, HCO3-Cl  it is not the first

R=221: for better understanding the water rock processes, should be very useful calculate the saturation index refers to main minerals in the aquifer

R=231: the other processes must also be discussed

R=241: indicate a reference on this value of 0.5!

R=269: regarding figure 7, what interpolation method was used, kriging, idw?

R=271: see comment on row 121

Discussion and conclusions should be revised taking into account the previous comments

ADD THESE REFERENCES

[1]-Fuoco, I., Marini, L., De Rosa, R., Figoli, A., Gabriele, B. and Apollaro, C., 2022. Use of reaction path modelling to investigate the evolution of water chemistry in shallow to deep crystalline aquifers with a special focus on fluoride. Science of The Total Environment, 830, p.154566.

[2]-Fuoco, I., Figoli, A., Criscuoli, A., Brozzo, G., De Rosa, R., Gabriele, B. and Apollaro, C., 2020. Geochemical modeling of chromium release in natural waters and treatment by RO/NF membrane processes. Chemosphere, 254, p.126696.

[3]- Coomar, P. and Mukherjee, A., 2021. Global geogenic groundwater pollution. In Global Groundwater (pp. 187-213). Elsevier.

[4]-  Apollaro, C., Tripodi, V., Vespasiano, G., De Rosa, R., Dotsika, E., Fuoco, I., Critelli, S. and Muto, F., 2019. Chemical, isotopic and geotectonic relations of the warm and cold waters of the Galatro and Antonimina thermal areas, southern Calabria, Italy. Marine and Petroleum Geology, 109, pp.469-483.

Author Response

(The authors gave the same response as above.)

Reviewer 3 Report

The reviewed manuscript is dealing with the Hydrochemical characteristics and formation mechanism of groundwater in Qingdao City, Shandong Province, China. The hydrochemical characteristics and water quality were assessed. The authors must work very hard to improve the quality of their manuscript for it to meet the strict requirements for publishing in Water.

1. Line 14; the causes of groundwater chemistry → the factors that influence groundwater chemistry.

2. Line 16; correlation analysis is a type of statistical analysis.

3. Line 25; Insert the introduction section's title.

4. Line 27; people[1-2] → people [1-2]. (here and onward Lines 38, 64,

5. Line 30; water quality. [3-4]. → water quality [3-4].

6. Lines 40:43; Please provide the reader with additional information about the groundwater situation in these regions.

7. Lines 44:49; Please revise this paragraph to avoid using the word "foreign" and to provide more information to the reader.

8. Lines 55:57; However, a further in-depth research is needed to understand the water resources situation of groundwater better and to use and protect groundwater resources reasonably. → However, additional in-depth research is required to better understand the groundwater resource situation and to use and protect groundwater resources sustainably.

9. Line 57; Some information about saltwater intrusion in coastal aquifers is required in the introduction section.

10. Line 68:72; Please revise the objectives (causes of groundwater?).

11. Lines 85:100; Please support this information with appreciate references.

12. Lines 87:100; Readers must work hard to understand the distinctions between the various types of groundwater mentioned (open rock-like pore water, clastic rock-like pore-fissure water, and bedrock 89 fissure water). It would be easier for readers if the authors presented this section with aquifer names (if any) and rearranged the groundwater situation as aquifer characteristics.

13. Figure 1; Please improve the quality of this figure (here and onward for Figures 4 and 7).

14. Lines 104:105; How did you differentiate between Quaternary pore water and bedrock fracture water?

15. Line 107; PH → pH

16. Line 108; H2SiO3-were → H2SiO3- were.  (What is use of this parameters?)

17. Lines 113:116; This part of the methodology should be expanded slightly to include information about the manufacturer of the used instruments, analytical quality control, accuracy, certified reference materials, and detection limits.

18. Figure 2; This figure should be combined with Figure 1.

19. Lines 120:123; Please revise this paragraph to avoid using the word "I" (here and onward in Line 271).

20. Lines 124:126; This part can be moved to the study objectives in the introduction section.

21. Line 134; groundwater. (i) The → groundwater: (i) The

22. Line 136; (ii)Groundwater → (ii) Groundwater

23. Line 138; Kou Wenjie [30] et al. → Kou Wenjie et al. [30].

24. Lines 140:143; Please revise this paragraph to avoid using the word "They".

25. Line 148; â‘¡Fave correction???

26. Table 1; Please revise CV values. For Example TDS (Min. 74.83; Max. 17138.1; CV 2.24) is not logical.

27. Lines 173:180; The TDS and major ion values in Table 1 show that there is no uniform distribution. Please revise the CV values as well as their interpretation. Please comment on the variation in the distribution of groundwater hydrochemical parameters in different water types (Fracture water, Pore groundwater, and Surface water).

28. Line 185; B and G.Incorporating → B and G. Incorporating

29. Line 186; Shukarev?? Please insert the appropriate citation.

30. Figure 4 and Lines 192:211; Groundwater chemistry should be based on both cations and anions, not just anions. Please revise this figure and its subsequent interpretations, and ensure that the units used (g/L) are consistent with table 1 (mg/L). Also, when discussing water quality, you should refer to standards (good water quality? What is the standard?).

31. Line 218; Mg2+, suggesting → Mg2+,suggesting

32. Lines 214:221; Please support your discussion with literature.

33. Lines 236:252; Please support your discussion with literature. Please see

Dieu, L.P.; Cong-Thi, D.; Segers, T.; Ho, H.H.; Nguyen, F.; Hermans, T. Groundwater Salinization and Freshening Processes in the Luy River Coastal Aquifer, Vietnam. Water 202214, 2358. https://doi.org/10.3390/w14152358

Saleh, A.; Gad, A.; Ahmed, A.; Arman, H.; Farhat, H.I. Groundwater Hydrochemical Characteristics and Water Quality in Egypt’s Central Eastern Desert. Water 202315, 971. https://doi.org/10.3390/w15050971

Abd El-Wahed, M.; El-Horiny, M.M.; Ashmawy, M.; El Kereem, S.A. Multivariate Statistical Analysis and Structural Sovereignty for Geochemical Assessment and Groundwater Prevalence in Bahariya Oasis, Western Desert, Egypt. Sustainability 202214, 6962. https://doi.org/10.3390/su14126962

34. Line 258; groundwater pollution → groundwater quality

35. Line 261; evaluation results s. → evaluation results.

36. Table 3; I, II, III, IV, and V classes should be defined in the Material and methods section. Please specify which samples (LS001, ..., ..., ...) correspond to each type of water (Fracture water, Pore groundwater, and Surface water).

37. Accordingly, the abstract and conclusion need to be revised.

38. The references style should be revised following the journal instructions.

Author Response

(The authors gave the same response as above.)

Round 2

Reviewer 1 Report

The manuscript is now much improved. The authors have done a great job of responding to all comments over the article. This manuscript will make a good contribution to the scientific community as a research article. Thank you for all your efforts. I recommend to accept this article to publish.

Author Response

We sincerely appreciate all of you for the time and effort that you have put into reviewing our manuscript entitled " Hydrochemical characteristics and formation mechanism of groundwater in Qingdao City, Shandong Province, China" (water-2285381). Your comments and suggestions are very important and helpful in improving our work.

Reviewer 2 Report

Remarks from reviewers have been done, and the paper is now more focuse on his core topic
Author Response

(The authors gave the same response as above.)

Reviewer 3 Report

The authors' revision improvements appear to be acceptable. Some issues remain to be addressed.

1. Figure 1: Please improve the quality of this figure (size and resolution).

2. Figure 2: Please improve the quality of this figure (size and resolution) and you should use a single figure-caption indication for the explanation for the sub-figures label (a-d). For Example Figure 2. (a)……..; (b)………..

3. Line 185; Please remove.

4. Table 1; Thank you for double-checking your data, however, there are still some issues. (Your answer: the coefficient of variation looks and feels not very reasonable). If the CV does not provide a clear explanation for the distribution of the studied parameters, another statistical parameter, such as the Kolmogorov-Smirnov (K-S) test (or any alternative), should be used.

5. Lines 258:260; You should refer to the used standard or criterion for classifying groundwater samples.

6. Line 262; Shukarev?? Please insert the appropriate citation.

7. You should support your discussion in all parts (sections 3.1 to 3.4) with the literature.

Author Response

Dear Editor and Reviewers,

We sincerely appreciate all of you for the time and effort that you have put into reviewing our manuscript entitled " Hydrochemical characteristics and formation mechanism of groundwater in Qingdao City, Shandong Province, China" (water-2285381). Your comments and suggestions are very important and helpful in improving our work. We have revised our manuscript carefully according to these comments and all of the revisions are highlighted in the “revised manuscript”. 
